

# Dietary fatty acid patterns and risk of oesophageal squamous cell carcinoma

Chanchan Hu[1], Zheng Lin[1], Zhiqiang Liu[1], Xuwei Tang[1], Jianyu Song[1], Jianbo Lin[2], Yuanmei Chen[3] and Zhijian Hu[1,4]

[1] Department of Epidemiology and Health Statistics, School of Public Health, Fujian Medical University, Fuzhou, Fujian, China
[2] Department of Thoracic Surgery, The First Affiliated Hospital of Fujian Medical University, FuZhou, Fujian, China
[3] Department of Thoracic Surgery, Fujian Medical University Cancer Hospital, FuZhou, Fujian, China
[4] Key Laboratory of Ministry of Education for Gastrointestinal Cancer, Fujian Medical University, FuZhou, Fujian, China

## ABSTRACT

**Background**. To characterize and examine the associations between dietary fatty acid intake patterns and the risk of oesophageal squamous cell carcinoma (ESCC).

**Methods**. A total of 422 patients and 423 controls were recruited. Dietary fatty acids were entered into a factor analysis. Multivariable logistic regression and restricted cubic spline were used to evaluate the risk of ESCC specific for different dietary fatty acid patterns (FAPs). A forest plot was applied to show the association between FAPs and ESCC risk after stratification by lifestyle exposure factors (tobacco smoking, alcohol drinking, pickled food, fried food, hot food, hard food).

**Results**. The factor analysis generated four major fatty acid patterns: a medium- and long-chain SFA (MLC-SFA) pattern; an even-chain unsaturated fatty acid (EC-UFA) pattern, a saturated fatty acid (SFA) pattern and an n-3 long-chain polyunsaturated fatty acid (n-3 LC-PUFA) pattern. In the multivariate-adjusted model, the odds ratios (ORs) with 95% confidence intervals (CIs) of ESCC were 2.07 (1.31, 3.26) and 0.53 (0.34, 0.81) for the highest versus the lowest tertiles of the EC-UFA pattern and n-3 LC-PUFA pattern, respectively. The MLC-SFA and SFA patterns were not associated with ESCC. An association between FAPs and ESCC risk after stratification by lifestyle exposure factors was also observed.

**Conclusions**. Our study indicates that the EC-UFA pattern and n-3 LC-PUFA pattern intake are associated with ESCC, providing a potential dietary intervention for ESCC prevention.

## INTRODUCTION

Oesophageal carcinoma has the 7th highest incidence and constitutes the 6th leading cause of cancer deaths worldwide by GLOBOCAN 2020 (*Sung et al., 2021*). Asia accounts for approximately 78% of all oesophageal carcinoma cases, while 49% of cases occur in China (*Uhlenhopp et al., 2020*). As the sixth-most common and fourth-most deadly cancer in China, the age-adjusted 5-year survival of oesophageal carcinoma is relatively poor, in

Corresponding author
Zhijian Hu, huzhijian@fjmu.edu.cn

the range of 20–30% (*Cao et al., 2021*; *Li et al., 2018*), and more than 90% of oesophageal cancer cases are ESCC (*Liang, Fan & Qiao, 2017*).

The incidence and prognosis of ESCC are affected by many factors. Tobacco smoking, alcohol consumption, and environmental carcinogen exposure are the major recognized risk factors for ESCC (*Huang et al., 2021*; *Thrift & Hepatology, 2021*). In recent years, a growing body of research has highlighted the key role of nutrition in this cancer (*Julibert et al., 2019*; *Santiago et al., 2018*). Among the nutritional factors, the role of fatty acids in tumorigenesis is becoming increasingly concerning. At present, some studies have found that fatty acids are closely related to the risk of various types of cancer, such as lung cancer (*Luu et al., 2018*), pancreatic cancer (*Qiu et al., 2020*), colorectal cancer (*Nguyen et al., 2021*), and oesophageal cancer (*Zamani et al., 2020*). However, both theoretically and empirically, due to the limitations of a single nutrient or food approach, dietary pattern assessment has become an alternative method for measuring dietary exposure in nutritional epidemiology (*Newby & Tucker, 2004*). FA pattern analysis demonstrates the interaction between various fatty acids and explains their complex association with diseases (*Choi, Ahn & Joung, 2020*; *Warensjö et al., 2006*). The combinations of multiple fatty acids may influence ESCC risk more than single fatty acids. Therefore, it is important to consider the intake of fatty acid patterns. To the best of our knowledge, the association of a combination of fatty acids with incident ESCC has not been evaluated. Therefore, we hypothesized that different dietary FAPs are associated with the risk of ESCC. The present study aimed to define specific dietary FAPs and investigate the relations between the generated FAPs and ESCC.

## MATERIALS & METHODS

### Study design and subjects

A hospital-based case–control study was conducted in Fujian Province, China. Patients were recruited from the Fujian Provincial Cancer Hospital and the First Affiliated Hospital of Fujian Medical University. The inclusion criteria for cases were as follows: (1) newly diagnosed primary patients who were histologically or cytologically diagnosed with ESCC (International Classification of Diseases 10th revision); (2) all cases were diagnosed with macroscopic type confirmed; and (3) Chinese Han population who resided in Fujian Province for at least the past 10 years. The exclusion criteria for cases were as follows: (1) patients with second primary, recurrent or metastasized cancer, and previous radiotherapy or chemotherapy were excluded; (2) those who had extreme daily caloric intake (>4,400 Kcal or <500 Kcal); and (3) those who did not accept or cooperate with the questionnaire. Finally, a total of 422 cases of ESCC were enrolled from Fujian Provincial Cancer Hospital (FPCH) (176 cases) and the First Affiliated Hospital of Fujian Medical University (246 cases) in Fuzhou City during the period from June 2014 through December 2020. Meanwhile, 423 controls were randomly chosen from community residents who ordered health examinations with nonneoplastic conditions. The control group inclusion criteria were as follows: (1) Chinese Han population resided in Fujian Province for at least the past 10 years and (2) controls with nonneoplastic conditions. The exclusion criteria of the

control group were (1) those who had extreme daily caloric intake (>4,400 Kcal or <500 Kcal) and (3) those who did not accept or cooperate with the questionnaire. This study was approved by the Institutional Review Boards of Fujian Medical University (NO. 2011052). Written informed consent was obtained from all participants before they participated in the study. All investigations performed in this study were conducted in accordance with the guidelines of the 1975 Declaration of Helsinki. A standard questionnaire was administered to cases and controls by specially trained interviewers. Questions covered demographic characteristics (*e.g.*, gender, age, education level, marital status), dietary habits, lifestyle habits such as tobacco smoking and alcohol drinking, personal medical history, family history of cancer, and dietary factors (*e.g.*, pickled food, fried food, hot food, hard food).

## Assessment of dietary intake

The usual diet was assessed by a food frequency questionnaire (FFQ). The contents of the questionnaire mainly included: (1) grain, 14 items; (2) Beans and their products, eight items; (3) Vegetables, 52 items; (4) Fruit, 30 items; (5) Animal food, 49 items; (6) Bacteria, algae, and nuts, eight items; (7) Beverages, drinks, and soups, 10 items. There are seven food categories, and the total number of food items is 171. For each food item, participants were asked, on average, how often they had eaten that food over the past year, based on the commonly used unit or portion size. The dietary data during the 1 year before the diagnosis for cases or the year before the interview for controls were selected. Food items from the FFQ are converted into dietary fatty acids. The intake of dietary fatty acids in food of each item = the intake of food of each item (g/d) × the content of dietary fatty acids in the edible part of the food (100 g)/100 g. The fatty acids and energy content reference to 2018 "China Food Composition Tables-Standard edition". To ensure that the data followed a normal distribution, natural logarithm (LN) conversion was carried out for the total energy and the daily intake of various dietary fatty acids, and then FAs intakes were adjusted for energy intake using the residuals method (*Willett & Stampfer, 1986*). Missing values in the fatty acid data were replaced by the variable mean.

## Definition of variables

Subjects who had smoked at least one cigarette per day for 6 months were considered tobacco smokers (*Steevens et al., 2010*). Subjects who had at least one drink per week for at least 6 months were alcohol drinkers (*Yang et al., 2017*). Tea drinkers were defined as dranking at least one cup of tea per week continuously for more than 6 months (*Wang et al., 2007*). Hot food intake means eating high-temperature processed meals within 5 min. Hard foods are defined as are harder to chew in the diet, such as peanuts, walnuts, chestnuts, beef jerky, etc. For dietary data, participants were asked how often intake each food item according to the following options: ≥5 times per week, 3–4 times per week, 1–2 times per week, less than one time per week, or not at all.

## Statistical analyses

After investigating all case and control subjects, the data were recorded using Epidata software (v3.1), with double-entry verification. The distributions of demographic characteristics, substance uses (tobacco, alcohol, and tea) of ESCC patients, and controls

were compared by the chi-square test. We used the Mann–Whitney $U$ test to test whether there was any difference in dietary fatty acid content distribution between the case and control groups. For descriptive purposes, we generated a hierarchical cluster tree to visually evaluate the correlations between individual fatty acids (*Hearty & Gibney, 2009*). Pearson correlation coefficients between fatty acids were also calculated.

A factor analysis (*Newby & Tucker, 2004*) based on energy-adjusted fatty acid intake values to derive dietary fatty acid patterns (FAPs). Oblique rotation was used to derive dietary FAPs to obtain a simpler structure with greater interpretability. Finally, four FAPs were extracted considering eigenvalue (>1), scree plot, and variance contribution. Participants were grouped into tertiles (T) according to the factor score of each pattern. The lowest score groups of each FAP were used as the reference. We used Spearman correlation analysis to assess the associations between each pattern score and the intake of 4 food groups (cereals, meat, freshwater fish, deep-sea fish) and 3 oils (peanut oil, animal oil, blend oil).

We applied three multivariable logistic regression models to estimate the OR value and 95% CI between the dietary fatty acid pattern score and ESCC risk. Model 1 adjusted for gender, age, education level, marital status, family tumour history, occupation, tobacco smoking, alcohol drinking, and tea consumption. To further control the influence of dietary habits on outcomes, hot food, hard food, pickled food, and fried food were included in Model 2. Model 3 adjusted for Model 2 and the other three FA scores. The fitting performance of the three models was evaluated by the Akaike information criterion (AIC). We used restricted cubic splines to visualize the trend of dietary fatty acid scores with the risk in ESCC.

Finally, we used a forest plot to demonstrate the association between FAPs and ESCC risk after stratification by lifestyle exposure factors (tobacco smoking, alcohol drinking, pickled food, fried food, hot food, hard food). The correlation between dietary fatty acid patterns and clinicopathological factors (T, N, M stages) was also explored. All analyses were performed using R 4.0.3 software, with $\alpha_{\text{two-sided}} = 0.05$.

## RESULTS

### General situation and comparison of dietary fatty acids

Table 1 presents demographic characteristics and lifestyle risk factors distribution in the patient groups and controls. Table S1 shows the difference in fatty acid intake between the case and control groups after energy adjustment.

### The fatty acid–factor loadings of the four major factors

The correlation matrix of 36 fatty acids is shown in Fig. 1. Factor analyses, including 36 major fatty acids, identified four factors that explained 60.8% of the variation in these variables in the study population. A similar pattern was identified in cluster analysis, as fatty acids adjacent to the tree had similar loading values (Fig. 1). Four FA factor scores were extracted to construct the FAP score of dietary fatty acids. Based on the major contributors to each pattern (Table 2). Pattern 1 components mainly included 20:0, 16:0, 11:0, 12:0, 8:0, 18:0, 17:0, 18:1 and 20:2, characterized by medium- and long-chain saturated fatty
**Table 1  Distribution of characteristics among cases and controls ($n = 845$).**

| Variable | Controls $n$ (%) | Cases $n$ (%) | $\chi^2$ | $P$ |
|---|---|---|---|---|
| Age (years) | | | 20.934 | <0.001 |
| ⩽54 | 248 (58.6%) | 181 (42.9%) | | |
| >54 | 175 (41.4%) | 241 (57.1%) | | |
| Gender | | | 0.057 | 0.812 |
| Male | 240 (56.7%) | 236 (55.9%) | | |
| Female | 183 (43.3%) | 186 (44.1%) | | |
| Education level | | | 81.055 | <0.001 |
| Low | 161 (38.1%) | 291 (69.0%) | | |
| High | 262 (61.9%) | 131 (31.0%) | | |
| Marital status | | | 11.252 | 0.001 |
| Married | 392 (92.7%) | 412 (97.6%) | | |
| Other | 31 (7.3%) | 10 (2.4%) | | |
| Occupation | | | 113.326 | <0.001 |
| Farmer and worker | 145 (34.3%) | 299 (70.9%) | | |
| Other | 278 (65.7%) | 123 (29.1%) | | |
| Income (RMB/monthly) | | | 27.944 | <0.001 |
| <2,000 | 302 (71.4%) | 364 (86.3%) | | |
| ⩾2,000 | 121 (28.6%) | 58 (13.7%) | | |
| Tobacco smoking | | | 31.699 | <0.001 |
| No | 291 (68.8%) | 210 (49.8%) | | |
| Yes | 132 (31.2%) | 212 (50.2%) | | |
| Alcohol drinking | | | 1.012 | 0.314 |
| No | 254 (60.0%) | 239 (56.6%) | | |
| Yes | 169 (40.0%) | 183 (43.4%) | | |
| Tea consumption | | | 9.279 | 0.002 |
| No | 148 (35.0%) | 191 (45.3%) | | |
| Yes | 275 (65.0%) | 231 (54.7%) | | |
| Family history of cancer | | | 0.035 | 0.852 |
| No | 364 (86.1%) | 365 (86.5%) | | |
| Yes | 59 (13.9%) | 57 (13.5%) | | |

acids (the "MLC-SFA" pattern). Pattern 2 was characterized by high positive loadings from 22:6, 24:1, 20:5, 20:1, 20:3, 20:4 and 24:0 in SFAs; which was called the "EC-UFA" pattern. We characterized the third FA pattern as an "SFA pattern", with high factor loading of 6:0. 10:0, 4:0, 13:0, 14:0 and 14:1. The fourth pattern was characterized by high positive loadings from 22:3, 22:5, 22:4 and 15:0, which we named the "n-3 LC-PUFA" pattern.

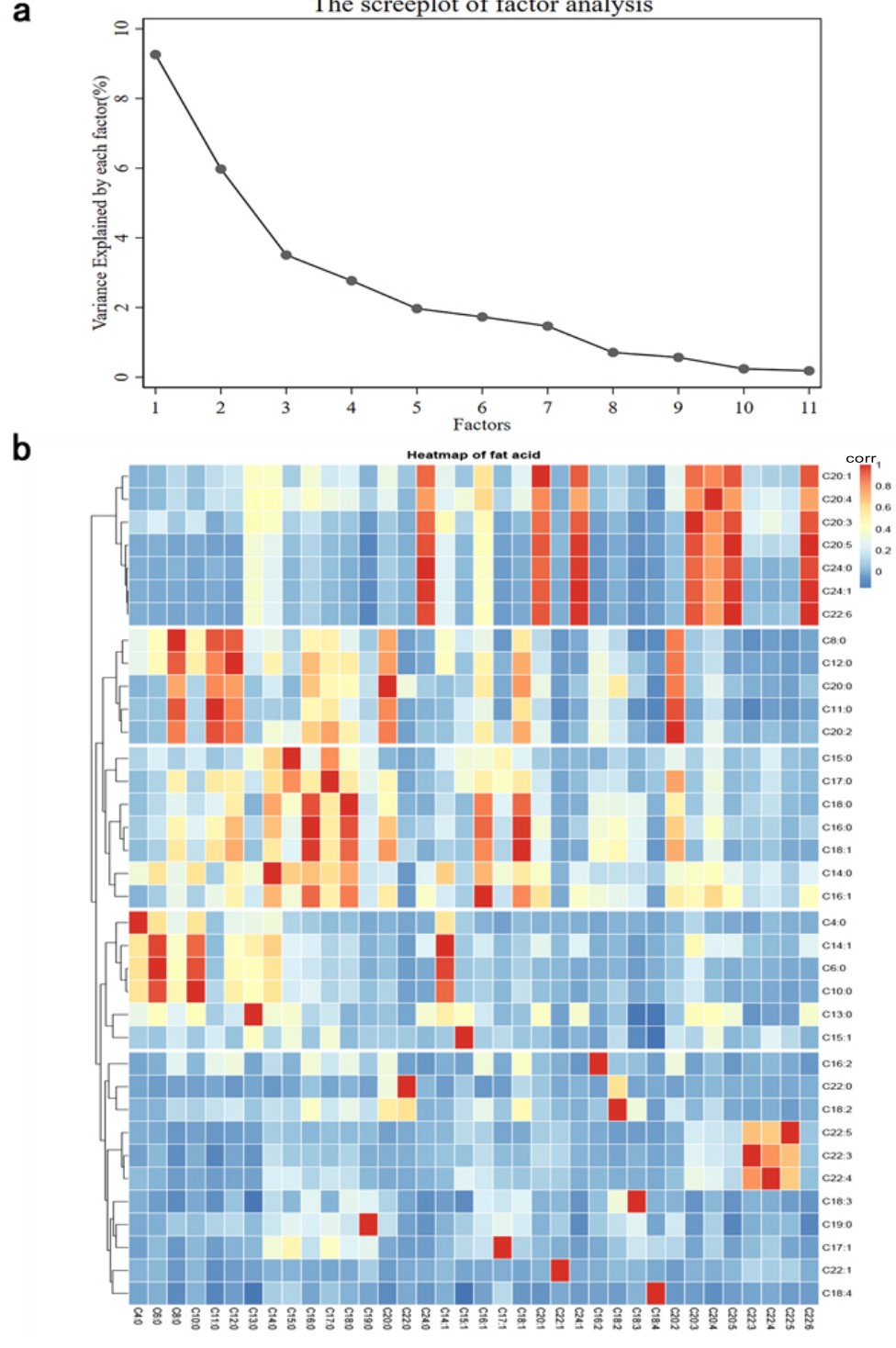

**Figure 1** **Principal components and clusters of 36 fatty acids.** (A) The proportion of total variance of 36 fatty acids explained by each principal component. (B) Hierarchical cluster tree on the left and the heatmap of fat acid on the right.

**Table 2** The fatty acidfactor loadings of the four major patterns.

| Fatty acid | Pattern 1 MLC-SFA | Pattern 2 EC-UFA | Pattern 3 SFA | Pattern 4 n-3 LC-PUFA |
|---|---|---|---|---|
| 18:1 | 0.924[a] | 0.001 | 0.008 | 0.101 |
| 20:2 | 0.914[a] | 0.026 | 0.028 | 0.136 |
| 20:0 | 0.898[a] | 0.078 | 0.058 | 0.247 |
| 16:0 | 0.835[a] | 0.115 | 0.075 | 0.199 |
| 11:0 | 0.820[a] | 0.029 | 0.061 | 0.386 |
| 12:0 | 0.816[a] | 0.063 | 0.409 | 0.226 |
| 8:0 | 0.723[a] | 0.048 | 0.422 | 0.375 |
| 18:0 | 0.701[a] | 0.007 | 0.037 | 0.394 |
| 17:0 | 0.664[a] | 0.005 | 0.117 | 0.316 |
| 16:1 | 0.656[a] | 0.425 | 0.023 | 0.312 |
| 16:2 | 0.449 | 0.053 | 0.059 | 0.032 |
| 18:2 | 0.435 | 0.017 | 0.08 | 0.057 |
| 22:0 | 0.131 | 0.038 | 0.129 | 0.049 |
| 22:6 | 0.084 | 0.993[a] | 0.028 | 0.051 |
| 24:1 | 0.075 | 0.990[a] | 0.026 | 0.089 |
| 20:5 | 0.092 | 0.988[a] | 0.033 | 0.028 |
| 24:0 | 0.054 | 0.968[a] | 0.03 | 0.086 |
| 20:1 | 0.174 | 0.932[a] | 0.018 | 0.015 |
| 20:3 | 0.078 | 0.928[a] | 0.18 | 0.112 |
| 20:4 | 0.218 | 0.809[a] | 0.008 | 0.207 |
| 6:0 | 0.005 | 0.063 | 0.961[a] | 0.004 |
| 10:0 | 0.081 | 0.076 | 0.932[a] | 0.038 |
| 14:1 | 0.018 | 0.188 | 0.916[a] | 0.024 |
| 4:0 | 0.038 | 0.055 | 0.714[a] | 0.043 |
| 13:0 | 0.002 | 0.384 | 0.528[a] | 0.036 |
| 14:0 | 0.383 | 0.209 | 0.526[a] | 0.465 |
| 22:4 | 0.025 | 0.106 | 0.042 | 0.708[a] |
| 22:3 | 0.069 | 0.122 | 0.068 | 0.647[a] |
| 22:5 | 0.09 | 0.115 | 0.067 | 0.566[a] |
| 15:0 | 0.229 | 0.003 | 0.276 | 0.55[a] |
| 17:1 | 0.129 | 0.072 | 0.031 | 0.47 |
| 19:0 | 0.264 | 0.104 | 0.015 | 0.265 |
| 18:3 | 0.196 | 0.081 | 0.127 | 0.239 |
| 15:1 | 0.128 | 0.068 | 0.144 | 0.207 |
| 22:1 | 0.06 | 0.004 | 0.015 | 0.169 |
| 18:4 | 0.057 | 0.075 | 0.021 | 0.146 |
| Eigen value | 9.312 | 6.020 | 3.626 | 2.930 |
| Total variance (%) | 25.865 | 16.721 | 20.071 | 8.139 |

**Notes.**
[a] Factor loadings that contribute to defining each factor.

**Table 3** The association between dietary fatty acid patterns and esophageal cancer.

| Model | Tertile of the fatty acid pattern score[a] | | | $P_{trend}$ | AIC |
|---|---|---|---|---|---|
| | I | II | III | | |
| PC1: MLC-SFA | | | | | |
| Case/control (n) | 141/140 | 112/171 | 169/112 | | |
| Model 1 | 1.0 (reference) | 0.648 (0.440,0.953) | 1.278 (0.867,1.833) | 0.201 | 997.614 |
| Model 2 | 1.0 (reference) | 0.716 (0.481,1.064) | 1.338 (0.900,1.989) | 0.147 | 962.2605 |
| Model 3 | 1.0 (reference) | 0.681 (0.448,1.033) | 1.309 (0.862,1.988) | 0.159 | 953.3695 |
| PC2: EC-UFA | | | | | |
| Case/control (n) | 114/167 | 150/133 | 158/123 | | |
| Model 1 | 1.0 (reference) | 1.849 (1.256,2.723) | 1.915 (1.285,2.856) | 0.001 | 995.8135 |
| Model 2 | 1.0 (reference) | 1.887 (1.264,2.817) | 2.079 (1.376,3.142) | <0.001 | 956.6683 |
| Model 3 | 1.0 (reference) | 1.694 (1.104,2.600) | 2.069 (1.314,3.257) | 0.002 | 952.2726 |
| PC3: SFA | | | | | |
| Case/control (n) | 151/130 | 151/132 | 120/161 | | |
| Model 1 | 1.0 (reference) | 1.176 (0.802,1.724) | 0.754 (0.513,1.108) | 0.154 | 1013.972 |
| Model 2 | 1.0 (reference) | 1.106 (0.746,1.640) | 0.720 (0.484,1.070) | 0.107 | 974.136 |
| Model 3 | 1.0 (reference) | 0.978 (0.655,1.459) | 0.607 (0.400,0.920) | 0.021 | 950.7284 |
| PC4: n-3 LC-PUFA | | | | | |
| Case/control (n) | 182/99 | 109/174 | 131/150 | | |
| Model 1 | 1.0 (reference) | 0.439 (0.297,0.650) | 0.517 (0.348,0.769) | 0.001 | 980.3383 |
| Model 2 | 1.0 (reference) | 0.453 (0.302,0.678) | 0.608 (0.401,0.921) | 0.017 | 943.6093 |
| Model 3 | 1.0 (reference) | 0.452 (0.299,0.682) | 0.525 (0.340,0.811) | 0.003 | 932.1294 |

**Notes.**

[a]Three categories were obtained by tertile of the fatty acid pattern score. Each participant was assigned a fatty acid pattern score. Ultivariable-adjusted Logistic regression models.
Model 1 adjusted for demographic characteristics: gender, age, education level, marital status, family history of cancer, occupation, tobacco smoking, alcohol drinking, tea consumption.
Model 2 adjusted for Model 1 and hard food, hot food, pickled food, fired food.
Model 3 adjusted for Model 2 and other three FA scores.

## Correlation between dietary fatty acid scores and food groups

The correlations of the four FACP scores with intake of food and oil groups are shown in Table S2. The MLC-SFA pattern score was positively correlated with the intake of "animal oil" ($r = 0.124$, $P = 0.002$). The EC-UFA pattern represented a diet relatively high in "peanut oil" ($r = 0.189$, $P < 0.001$). The "n-3 LC-PUFA pattern" score was positively correlated with the intake of "deep-sea fish" ($r = 0.099$, $P = 0.015$).

## Association of fatty acid pattern score with ESCC incidence

The model fit performance was evaluated according to the AIC, and Model 3 had the lowest AIC and the best fitting effect. In multivariable analyses, the n-3 LC-PUFA pattern was associated with a lower likelihood of ESCC after adjustment for all covariates (OR: 0.53, 95% CI [0.34, 0.81], $P = 0.003$). After adjustment for potential confounding variables, the EC-UFA pattern was positively associated with ESCC (OR: 2.07, 95% CI [1.31, 3.26], $P = 0.002$). The MLC-SFA pattern and SFA pattern were not significantly associated with ESCC. (Table 3).

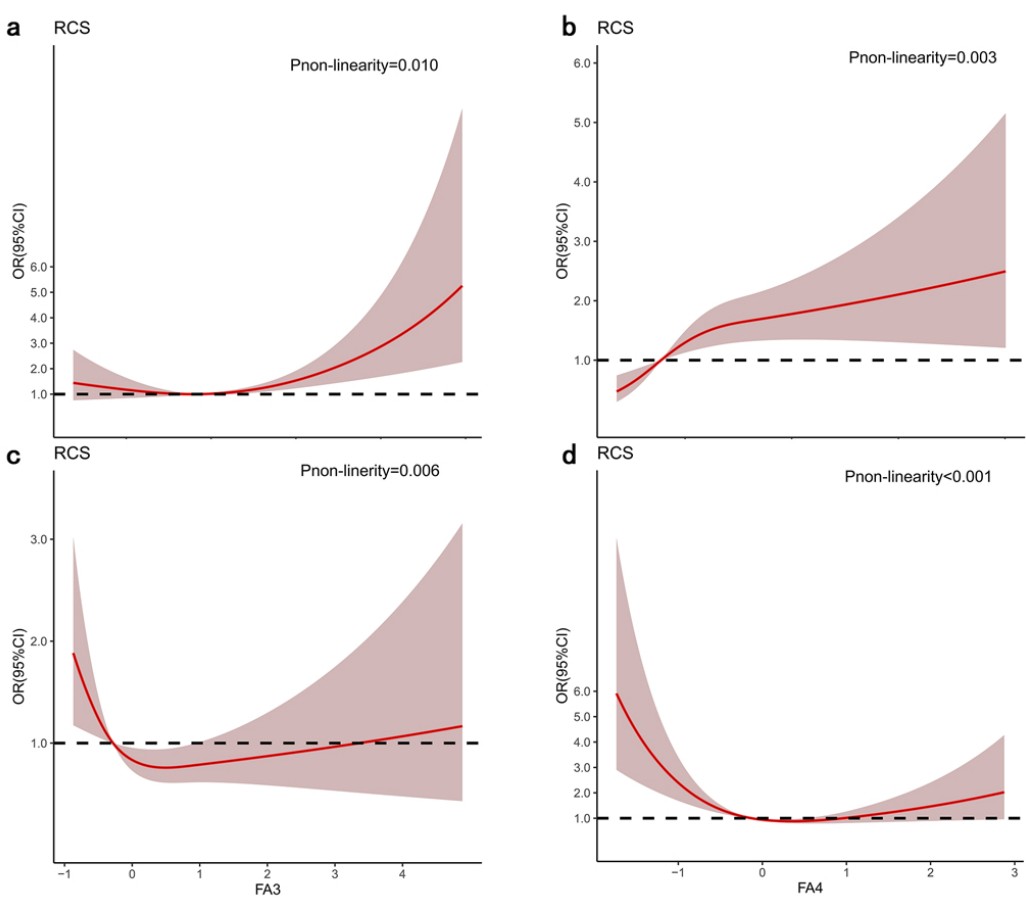

**Figure 2   Odds radios (ORs) based on four FAPs and restricted cubic splines.** (A) MLC-SFA pattern; (B) EC-UFA pattern; (C) SFA pattern; (D) n-3 LC-PUFA pattern. The shaded areas indicate the 95% confidence intervals of ORs. The result adjusted for model 3.

## Linear trend of dietary fatty acid score and incidence of ESCC

The dose–response relationship between the intake of the four FAPs and the risk of ESCC is shown in Fig. 2. There was a nonlinear positive association between the EC-UFA pattern and the risk of ESCC ($p$ for nonlinearity <0.05). Nevertheless, there was a nonlinear negative association between the n-3 LC-PUFA pattern and the risk of ESCC ($p$ for nonlinearity <0.001) (Fig. 2).

## Stratification analysis between dietary fatty acid patterns and the risk of ESCC

Figure 3 shows that the associations between FAPs and ESCC risk were stratified by lifestyle exposure factors. When stratified by fried foods, the association between the MLC-SFA pattern and ESCC risk was different ($I^2 = 75.7\%$, $P_{\text{heterogeneity}} = 0.043$). The association between the EC-UFA pattern and ESCC risk varied across pickled foods ($I^2 = 78.2\%$, $P_{\text{heterogeneity}} = 0.032$). When stratified by tobacco smoking, the correlation between the SFA pattern and ESCC risk has changed ($I^2 = 99.3\%$, $P_{\text{heterogeneity}}<0.001$). Moreover,

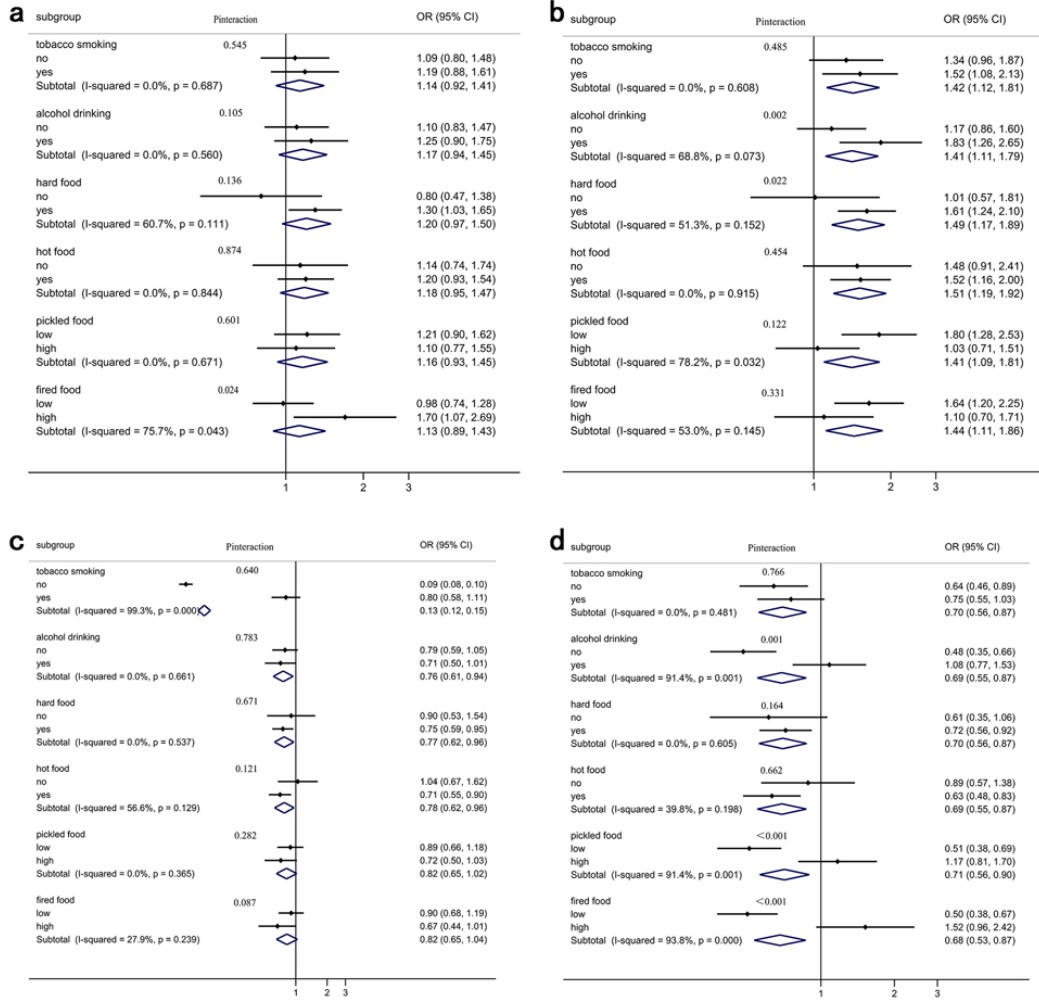

**Figure 3** **Four FAPs: dietary fatty acid pattern score and the risk of ESCC stratified by lifestyle exposure factors.** (A) MLC-SFA pattern; (B) EC-UFA pattern; (C) SFA pattern; (D) n-3 LC-PUFA pattern.

the associations were observed to be different between the n-3 LC-PUFA pattern and ESCC risk among those who consumed alcohol ($I^2 = 91.4\%$, $P_{\text{heterogeneity}} = 0.001$), pickled food ($I^2 = 91.4\%$, $P_{\text{heterogeneity}} = 0.001$), and fried food ($I^2 = 93.8\%$, $P_{\text{heterogeneity}} < 0.001$). Additionally, there were significant multiplicative interactions between the n-3 LC-PUFA pattern and ESCC risk across alcohol drinking ($P_{\text{interaction}} = 0.001$), pickled food ($P_{\text{interaction}} < 0.001$), and fried food ($P_{\text{interaction}} < 0.001$).

## Correlation between dietary fatty acid patterns and clinical pathological factors

Of the clinical pathological characteristics, the N stage significantly correlated with the EC-SFA pattern ($r_s = 0.270$, $P < 0.001$). The higher intake of EC-UFA patterns might be associated with a higher risk of advanced disease. M stage was negatively correlated with the

pattern of n-3 LC-PUFAs ($r_s = -0.175$, $P = 0.003$). However, T stage had no significant correlation with FAPs (Table S3).

# DISCUSSION

In this hospital-based case–control study, four main dietary patterns were identified, *i.e.,* medium- and long-chain SFA (MLC-SFA), even-chain unsaturated fatty acid (EC-UFA), SFA, and n-3 long-chain polyunsaturated fatty acid (n-3 LC-PUFA) patterns. The EC-UFA pattern was associated with an increased risk of ESCC, whereas the n-3 LC-PUFA pattern was associated with a decreased risk. However, there was no significant association with the MLC-SFA or SFA patterns observed in the study subjects.

Although nutrition has consistently been an important determinant of ESCC risk, the impact of fatty acid intake on the aetiology of ESCC has not been thoroughly investigated, especially in high-risk countries. In addition, only a few studies have evaluated the cumulative effect of fatty acid intake on ESCC risk using factor analysis (principal component) (*Bravi et al., 2012*; *De Stefani et al., 2008*), while factor analysis as a posteriori method allows the study of synergy among nutrients and the consequences of the interactions between them (*Jacobs & Steffen, 2003*). To our knowledge, there has been no attempt to assess the effect of dietary FAPs on ESCC risk. Still, some components in the dietary patterns of this study were similar to those of the patterns defined in other studies, although not all components were identical. The "SFA pattern" and "n-3 LC-PUFA pattern" were similar to the FAPs from South Africa (*Ojwang et al., 2019*), Korea (*Choi, Ahn & Joung, 2020*), and Uppsala (*Warensjö et al., 2006*).

The "MLC-SFA pattern" and "SFA pattern" were not significantly associated with the risk of ESCC. SFAs are major sources of energy (*Schoenfeld & Wojtczak, 2016*) and have strong antibacterial effects (*Yoon et al., 2018*), but there is a lack of evidence for the effect of SFAs on ESCC risk. In a case–control study with adults in Iran, higher levels of SFAs were associated with a lower risk of ESCC (*Hajizadeh et al., 2012*). However, in a study in South Africa, the "SFA pattern" was positively associated with measures of adiposity and metabolic syndrome (*Ojwang et al., 2019*). There was no evidence that MLC-SFA and SFA patterns were associated with ESCC risk in our study.

However, those adhering more to the "even-chain UFA pattern" were found to be at a higher risk of ESCC. The "even-chain UFA pattern" had a high factor loading of even-chain UFAs such as DHA (C22:6), nervonic acid (C24:1), EPA (C20:5), eicosenoic acid (C20:1), eicosatrienoic acid (C20:3), and AA (C20:4). Studies have shown that dietary unsaturated fatty acids are associated with an increased risk of cancer (*Vinciguerra et al., 2009*). In particular, AA in the even chain fatty acid pattern is a precursor to proinflammatory molecules (*Bojková, Winklewski & Wszedybyl-Winklewska, 2020*). In humans, even-chain UFAs were correlated with composite inflammation measures and may thus influence the risk of cancer (*Chiu et al., 2018*). Inflammation is a crux of developing many chronic diseases, including cancer (*Snodgrass et al., 2013*). An inflammatory microenvironment is an important part of the tumour microenvironment (*Khandia & Munjal, 2020*). Chronic inflammation is the cause of tumour transformation (*Singh et al., 2019*). More studies are essential for exploring their association.

The last pattern of n-3 LC-PUFAs was characterized by a higher intake of DPA (22:5) and docosatrienoic acid (22:3). Many previous studies have proven that the n-3 series of unsaturated fatty acids are mainly derived from fish (*Hidaka et al., 2015*). Our study also found a positive correlation between the fourth FAP and the intake of deep-sea fish. The n-3 LC-PUFA pattern is an important fatty acid that may play a role in preventing some cancers (*Jacobs & Steffen, 2003*). Furthermore, the n-3 LC-PUFA pattern, which has pleiotropic effects and enhances cancer cell apoptosis, modulates various eicosanoid pathways, leading to reduced inflammation, such as suppressing cyclooxygenase-2 synthesis inhibiting arachidonic acid-derived eicosanoids (*Mauermann, Pouliot & Fradet, 2011*). Animal studies and human observational studies have demonstrated that the n-3 LC-PUFA pattern may reduce the risk of breast, colon, and prostate cancers (*Mauermann, Pouliot & Fradet, 2011*; *VanderSluis et al., 2017*; *Yang et al., 2014*). In this study, the n-3 LC-PUFA pattern was also found to reduce the risk of ESCC.

When exploring the linear relationship between dietary fatty acid patterns and the risk of ESCC, we found that the risk of ESCC increased with increasing EC-UFA intake. However, as the intake of n-3 LC-PUFAs improved, the risk of ESCC decreased. They were associated with a dose–response risk of ESCC (Fig. 2). N-3 polyunsaturated fatty acids (PUFAs) express anti-inflammatory properties and prevent tumour progression (*Zamani et al., 2020*), similar to our result.

After stratifying by life exposure factors, we found that the relationship between EC-UFA pattern, n-3 LC-PUFA pattern and ESCC risk was modified by drinking, hard food, pickled food, and fried food. Fatty acids are variable with alcohol consumed, they can be either anti- or proinflammatory in nature (*Zirnheld et al., 2019*). This study has shown that alcohol consumption increased the risk of EC-UFA pattern for ESCC, and it may be due to the pro-inflammatory effects of EC-UFA pattern being exacerbated by alcohol. N-3 long-chain polyunsaturated fatty acids have anti-inflammatory effects. Our results suggest that n-3 LC-PUFA pattern has a protective effect on ESCC in non-drinking populations. However, in those who consumed alcohol, the protective effect was reversed to a dangerous effect, possibly because alcohol intake altered the fatty acid metabolic profile in vivo. Mechanical damage of hard food eating may impair the barrier function of the oesophageal epithelium, making it more vulnerable to inflammatory attack by EC-UFA. Pickled food is often preserved with the addition of nitrates or nitrites, which increases the formation of N-nitroso compounds (NOCs), considered animal carcinogens and possible human carcinogens (*Islami et al., 2009*). In this study, the protective effect of the n-3 LC-PUFA pattern was lost with the intake of pickled foods, presumably due to the high concentrations of salt directly damaging the oesophageal mucosa (*Onuk, Oztopuz & Memik, 2002*), leading to susceptibility to esophagitis and an increased risk of ESCC. There have been reports of a significant dose–response relationship between the intake frequency of fried food and the risk of ESCC (*Lin et al., 2011*). Cooking meat at high temperatures produces large amounts of polycyclic aromatic hydrocarbons and high levels of heterocyclic amines (*Knize et al., 1999*). Both groups of chemicals have been suggested to increase the risk of ESCC (*Okaru et al., 2018*; *Ward et al., 1997*). After stratification by fried foods, we found that the protective effect of the n-3 LC-PUFA pattern was weakened in those who

regularly consumed fried foods, and it is possible that the role of n3 fatty acids was altered by carcinogens. Its exact mechanism requires further investigations.

This is the first study to reveal the relationship between dietary FAPs and ESCC risk in a Chinese population. In our daily lives, people eat a diet made up of various fatty acids, not just one kind of fatty acid. Therefore, it is important to consider FAP analysis because it can reflect the actual dietary quality and summarize the effects of various dietary FAs. Compared with the traditional approach of analysing a single FA, factor analysis allows investigating the relationship between dietary components.

However, several limitations should be acknowledged in our study. Selection bias may exist in any hospital-based case–control study. However, all subjects were recruited from two hospitals according to strict criteria, minimizing selection bias. The study data were obtained from interviews and might lead to recall bias, which may limit the accuracy of our results. To alleviate this effect, we performed face-to-face interviews and defined variables. Finally, our cases were from patients with ESCC in Fujian Province, with a regional limitation. The control group was also from the Fujian Province. Therefore, this has no significant impact on the conclusion.

## CONCLUSIONS

In this study, we found that higher dietary intake of EC-UFA patterns was associated with a higher risk of developing ESCC. In contrast, a combination of individual fatty acids, characterized by an n-3 LC-PUFA pattern, was associated with a lower incidence of ESCC. Further prospective studies in various regions with larger sample sizes are needed to confirm this association.

## ACKNOWLEDGEMENTS

We thank the doctors, nurses, laboratory staff, and study participants for their contributions.

### Funding

This work was supported by the National Key R&D Program of China (No.2017YFC0907100), Medical Innovation project of Fujian Province (No.2018-CX-38) and the Startup Fund for scientific research, Fujian Medical University (No. 2018QH2012). The funders had no role in study design, data collection and analysis, decision to publish, or preparation of the manuscript.

### Grant Disclosures

The following grant information was disclosed by the authors:
National Key R&D Program of China: No. 2017YFC0907100.
Medical Innovation project of Fujian Province: No. 2018-CX-38.
Fujian Medical University: No. 2018QH2012.

## Competing Interests

The authors declare there are no competing interests.

## Author Contributions

- Chanchan Hu, Zheng Lin and Zhiqiang Liu conceived and designed the experiments, performed the experiments, analyzed the data, prepared figures and/or tables, authored or reviewed drafts of the paper, and approved the final draft.
- Xuwei Tang conceived and designed the experiments, prepared figures and/or tables, and approved the final draft.
- Jianyu Song conceived and designed the experiments, performed the experiments, prepared figures and/or tables, and approved the final draft.
- Jianbo Lin and Yuanmei Chen conceived and designed the experiments, authored or reviewed drafts of the paper, and approved the final draft.
- Zhijian Hu conceived and designed the experiments, analyzed the data, prepared figures and/or tables, authored or reviewed drafts of the paper, and approved the final draft.

## Human Ethics

The following information was supplied relating to ethical approvals (i.e., approving body and any reference numbers):

The study was approved by the Institutional Review Board of Fujian Medical University (number: 2011052).

## Data Availability

The raw data are available in the Supplemental File.

## Supplemental Information

Supplemental information for this article can be found online at http://dx.doi.org/10.7717/peerj.13036#supplemental-information.

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
