# Peer review of "Dietary fatty acid patterns and risk of oesophageal squamous cell carcinoma"

_PeerJ, doi:10.7717/peerj.13036_

## Round 0.1 · original submission · Major Revisions

How about the correlation between dietary fatty acid scores and clinical pathological factors including T, N, M stages of these ESCC patients?

·

Basic reporting

While the English overall is OK there are some bits that are not clear and it is not always clear whether this is a problem with English or something else.
The Abstract methods mention using a Forest plot to show effect modification, but you have not said what effect modifiers are to be studied.
What is ‘hard’ food? This is not meaningful to an international audience.
The first sentence of the introduction is poorly written. Should be incidence rather that incident and should give a reference to the source of the data as well as mentioning GLOBOCAN 2018.
I did not assess whether any key references were missing but think the following is misquoted. Line 208, ref 27 is specifically about MCFA not SFA and it is for MCFA that in animals attenuate weight gain.
There are no hypotheses provided.
Data is shared.
There is a problem with the reference list as most refs don’t have author names, just initials.
The legend for Fig 2 talks about error bars but there are no error bars on the figures, just shaded areas to represent 95% CI.

Experimental design

The introduction does not well justify why there might be an association between dietary fat intake and ESCC. If it has not been explored then use evidence from other studies to support why this might be important.
Why is multinomial logistic regression used? Isn’t the outcome ESCC yes/no?
In Abstract results there are too many significant figures in ORs and 95% CIs.
More information on the control selection is required. Were there any exclusion criteria applied to controls?
Was a sample size calculation performed?
More information on how FFQ responses were converted to nutrient intakes is required. How were serve sizes determined?
Why do you use percentage energy from the different fatty acids for the dietary pattern analysis when you have calculated the energy-adjusted fatty acid intakes using the residual method?
Line 144-5, we still don’t know what effect-modifiers are being considered and presented in Forest plots.
Why did you choose to dichotomise age at 54 years as shown in Table 1?
In Table 2 how did you decide which fatty acids defined each factor? It was clearly not just based on the magnitude of the loading alone.

Validity of the findings

The section headed ‘Cross-stratified heterogeneity test between dietary fatty acid score and ESCC incidence’ is very difficult to follow. Describe in words what all this actually means.
Line 267-268, even if you accept that the associations between dietary fatty acid patterns and ESCC in this case-control study are causal, factor analysis did not reveal the association between dietary FAPs and ESCC risk. Factor analysis just identified some patterns regarding how fatty acid intakes correlated with one another.

Additional comments

No comment

Reviewer 2 ·

Basic reporting

This is a novel topic, and it addresses the relation between the dietary fatty acid pattern and the risk of ESCC.

Experimental design

This manuscript is well organized and has an appropriate sample size.
The statistics for the analysis were clear and well described.
The inclusion criteria for the study were clearly stated in the article as well.

Validity of the findings

The authors made broader and more generalized conclusions based on the data than specified in the methods. The inclusion criteria stated that only the “Chinese Han population resided in Fujian Province” which differed from the conclusion.

Please include a paragraph or several sentences describing the limitation of the inclusion criteria in the discussion section.

Additional comments

Please improve the quality of the English in the manuscript.

1. Follow the general English writing rules, such as using a space after a punctuation mark.
2. There are some grammar mistakes, such as on lines 236-237 where the sentence cuts off abruptly.
3. Please only use English in the manuscript (line 315 need to be deleted).
4. Please use consistent formatting for references in the article and do not switch in between.
5. Adjust the font type of the manuscript to stay constant throughout and enlarge the font size of references.

---

## Round 0.2 · Minor Revisions

Some minor problems should be noticed as the reviewer 1 proposed.

·

Basic reporting

The authors have improved the manuscript but still some basic problems with the quality:
SFA is not defined in abstract.
Introduction: Spaces missing between text and brackets. This is a problem throughout the manuscript.
Line 44 oesophageal carcinoma is 6th leading cause of death globally but line 47 it is 4th most deadly cancer. Does this mean 4th leading cause of death? Is this for Asia rather than globally? Need to be clear.
Line 53, remove word ‘remarkably’ this is not appropriate here.
Please check carefully line 192-3, seems to be some confusion between pickled and fried foods.
The sentence lines 259-60 is not complete.
There are still some problems with the reference list. For example 329 Hajizadeh B, Jessri M, Akhoondan M, Moasheri S, and Rashidkhani BJDoteojotISfDotE. 2012. Nutrient patterns 330 and risk of esophageal squamous cell carcinoma: a case-control study. 25:442-448. 10.1111/j.1442- 331 2050.2011.01272.x
Check manuscript for typo where ‘fired food’ mixed up with ‘fried food’.
Line 166, what do you mean by the ‘most powerful’ factor?

Experimental design

In general the experimental design is appropriate but the section in the discussion that relates to the effect modification analysis is unconvincing, seems to be looking at additional risk factors for ESCC but not why they might modify effects of FAPs and this part of the analysis is not justified in methods. I would still like to see some justification for the stratification by hard foods and other variables in methods. Are they ESCC risk factors? Why would they modify the associations between FAPs and ESCC?
Line 133, you mean fatty acid intakes were energy-adjusted by the residual method (Willett ref) before the factor analysis?

Validity of the findings

It is reasonable to assume that the conclusion is valid for the population in your study but a limitation is that the results may not be applicable to other populations with different dietary habits and lifestyles. Given that China is a country with a high risk of ESCC the findings are unlikely to be relevant to countries with lower rates.

Additional comments

In general the authors have responded appropriately to reviewer's comments but the manuscript is still not at a publishable quality.

Reviewer 2 ·

Basic reporting

The authors have addressed all my concerns.

Experimental design

The authors have addressed all my concerns.

Validity of the findings

The authors have addressed all my concerns.

---

## Round 0.3 · accepted · Accept

The authors have replied to the reviewer's comments point by point, and the paper is acceptable.